# Ultrasonography Measurement of Renal Dimension and Its Correlation with Age, Body Indices, and eGFR in Type 1 Diabetes Mellitus Patients: Real World Data in Taiwan

**DOI:** 10.3390/jcm12031109

**Published:** 2023-01-31

**Authors:** Hsuan-An Su, Jung-Fu Chen, Chung-Ming Fu, Yueh-Ting Lee, Yi Wang, Chiang-Chi Huang, Jin-Bor Chen, Chien-Te Lee, Chien-Hsing Wu

**Affiliations:** 1Department of Dermatology, Far-Eastern Memorial Hospital, New Taipei City 22016, Taiwan; 2Division of Metabolism and Endocrinology, Department of Internal Medicine, Kaohsiung Chang Gung Memorial Hospital, Chang Gung University College of Medicine, Kaohsiung 83301, Taiwan; 3Division of Nephrology, Department of Internal Medicine, Kaohsiung Chang Gung Memorial Hospital, Kaohsiung 83301, Taiwan

**Keywords:** type 1 diabetes mellitus, ultrasonography, right renal length

## Abstract

Background: Assessment of renal size is clinically significant for the screening, diagnosis, and follow-up of renal diseases as the basis of clinical decisions. However, the relationship of renal dimension with age, body indices, and the estimated glomerular filtration rate (eGFR) has rarely been reported in the Chinese type 1 diabetes mellitus (T1DM) population. Methods: A total of 220 T1DM patients were retrospectively analyzed from the Chang Gung Research Database in Taiwan. Demographic data, laboratory data, and ultrasonographic images from January 2001 to November 2018 were extracted. Results: Eighty-five participants (38.6%) were male. The mean age was 34.2 years. The median eGFR was 60.0 mL/min/1.73 m^2^. The mean ultrasonographic left and right renal lengths (LL and RL) with S.D. were 10.9 ± 1.5 cm and 11.0 ± 1.1 cm, respectively. Renal lengths were longer with increasing body height and body weight but shorter with increasing age in patients with T1DM. In trajectory analysis, a linear mixed model revealed no significant trend in the changes in eGFR during the follow-up period. Moreover, renal length did not play a significant role in predicting KDIGO CKD stage 5 in the cohort. Conclusions: Renal length and its comparison to the reference ranges demonstrated very limited advantages in predicting renal function decline in T1DM patients.

## 1. Introduction

Type 1 diabetes mellitus (T1DM) is an autoimmune disease in which pancreatic β cells are compromised, resulting in insulin deficiency and hyperglycemia [1,2]. The incidence of T1DM is estimated as 22.9 cases per 100,000 people [3], and it has been increasing rapidly worldwide over the past decades, with an average annual increase of 3–4% [1]. Complications of T1DM involve multiple organs and can be life-threatening. Diabetic kidney disease (DKD) is a common T1DM complication that causes the progressive loss of renal function and is the leading cause of chronic kidney disease (CKD), followed by kidney failure (KF), requiring renal replacement therapy, and finally, premature mortality [4]. Although intensive glycemic control reduces the incidence of microvascular and macrovascular complications, most patients with T1DM still develop these complications [5]. In a 17-year Taiwanese population-based cohort study, renal diseases accounted for 11.45% of the causes of death in the T1DM population, and the standardized mortality ratio was 14.48, compared to the general population [6]. Early recognition of DKD in T1DM is crucial to delay the deterioration of renal function and to improve the outcomes of long-term, irreversible complications [4]. Renal ultrasonography is a useful tool for evaluating the renal morphology. Previously, we reported reference ranges for ultrasonographic renal dimensions in a large, healthy Taiwanese population, which was not only practical for screening for potential renal diseases in healthy subjects but also provided references for diseased populations [7]. Furthermore, renal ultrasonography measurements correlate with age, body height (BH), body weight (BW), and body mass index (BMI) [7]. However, the correlations between renal morphology and age or body indices have not yet been validated in a large T1DM population [8,9]. To date, studies that attempt to find correlations between renal ultrasonography measurements and the prediction of T1DM renal function decline are limited in literature, and data of renal function prediction in T1DM using ultrasonography are scarce. In the present study, we aim to delineate the ultrasonographic profile of renal dimensions in a large T1DM population in Taiwan and explore its potential correlations with renal function. Advanced understanding of ultrasonographic renal measurements in T1DM may provide implications for the early diagnosis of DKD and for the prediction of renal function decline.

## 2. Materials and Methods

### 2.1. Data Source and Study Population

We conducted a retrospective analysis at Kaohsiung Chang Gung Memorial Hospital using de-identified data retrieved from the Chang Gung Research Database (CGRD). CGRD is a validated electronic health record database established by the Chang Gung Medical Foundation, the largest medical system in Taiwan, and consists of data from all branches of Chang Gung Memorial Hospital, including the Keelung, Taipei, Linkou, Taoyuan, Chiayi, and Kaohsiung branches [10]. Demographic data, clinical diagnosis, medical records, laboratory data, and ultrasonographic images from January 2001 to November 2018 were extracted. The extraction protocol (Figure 1) was approved by the Institutional Review Board of the Chang Gung Medical Foundation, in accordance with the Helsinki Declaration of 1975 (IRB No. 201802374B0 and 201802374B0C601).The validation of the diagnoses of T1DM of the subjects was confirmed using the ICD-9 CM codes (25001, 25003, 25011, 25013, 25021, 25023, 25031, 25033, 25041, 25043, 25051, 25053, 25061, 25063, 25071, 25073, 25081, 25083, 26091, and 25093) or the ICD-10-CM codes (E10, E101, E1010, E1011, E102, E1021, E1022, E1029, E103, E1031, E10311, E10319, E1032, E10321, E10329, E1033, E10331, E10339, E1034, E10349, E1035, E10351, E10359, E1036, E1039, E104, E1040, E1041, E1042, E1043, E1044, E1049, E105, E1051, E1052, E1059, E106, E1061, E10610, E10618, E1062, E10620, E10621, E10622, E10628, E1063, E10630, E10638, E1064, E10641, E10649, E1065, E1069, E108, E109, O240, and O2401). In all branches of the Chang Gung Memorial Hospital, patients with suspected T1DM were referred to endocrinologists for diagnosis and treatment. Diagnostic information, including examination results, fasting or glucagon-stimulated C-peptide level, anti-glutamic acid decarboxylase antibody level, and history of diabetic ketoacidosis were sent to the National Health Insurance Administration for the approval of welfare and insurance. Furthermore, the ambulatory care expenditures from visits in medical records were reviewed to ensure that all the recruited subjects were provided with free blood glucose test strips, which were covered by national health insurance only for patients who met the diagnostic criteria of T1DM in Taiwan.

Clinical and demographic data were obtained within three months of their renal ultrasonography, including age, sex, BH, BW, BMI, serum creatinine level, and estimated glomerular filtration rate (eGFR). eGFR was derived from the Modification of Diet in Renal Disease formula. An average eGFR was adopted when there were more than two eGFR records within three months of renal ultrasonography. All renal ultrasonography examinations were performed by board-certified nephrologists. Kidney length was measured at the longest axis of the kidney, which was determined by several measurements taken from the sagittal or coronal plane or the planes in between. These data were used in this study without any reinterpretation or modification. The unit of measurement was centimeters (cm).

### 2.2. Statistical Analysis

The extracted data were organized using Microsoft Excel software and analyzed using SAS (version 9.4 SAS Institute, Cary, NC, USA). Descriptive statistics, Student’s *t*-test, nonparametric Mann–Whitney U test, and Spearman’s correlation analysis were performed to evaluate differences between continuous variables. Spearman’s correlation analysis was conducted to assess the relationships between right renal length (RL) and age, BH, BW, and eGFR. In trajectory analysis, eGFR data were obtained from the eGFR closest to the ultrasonographic measurement, the last eGFR data, or the first Kidney Disease Improving Global Outcomes (KDIGO) stage 5. Only subjects with more than one eGFR data obtained were included in the trajectory analysis. The best-fitting trajectory model was selected using the minimal Bayesian information criteria [11]. A linear mixed model was used to examine the trends of the trajectory models [12]. A survival analysis using the Cox proportional-hazards model was performed to determine whether renal length could be a risk factor for KDIGO CKD stage 5 in the subgroups derived from trajectory analysis. The effects of RL were calculated as crude hazard ratios with 95% confidence intervals. Statistical significance was set at *p* < 0.05.

## 3. Results

### 3.1. Subjects Characteristics

The demographic data of the 220 participants are shown in Table 1. Eighty-five (38.6%) patients were male. The mean age was 34.2 years. Mean BH, BW, and BMI with standard deviation (S.D.) were 160.7 ± 8.3 cm, 61.1 ± 11.9 kg, and 23.6 ± 3.9 kg/m^2^, respectively. The median eGFR was 60.0 mL/min/1.73 m^2^, with an interquartile range of 22–84.5 mL/min/1.73 m^2^. The mean ultrasonographic left renal length (LL) and RL with S.D. were 10.9 ± 1.5 cm and 11.0 ± 1.1 cm, respectively.

### 3.2. Correlations between Right Renal Length and the Body Indices in the T1DM Cohort

The line charts demonstrating the distribution of RL at different ages and the BH and BW groups with error bars indicating 95% confidence intervals are plotted in Figure 2A–C, where RL is longer with increasing BH and BW but shorter with increasing age.

The Spearman’s correlation coefficients (r) of the RL and body indices are listed in Table 2 and illustrated in Figure 3. RL was linearly correlated to age (r = −0.2268, *p* < 0.001), BH (r = 0.1835, *p* = 0.006), BW (r = 0.1717, *p* = 0.011), and eGFR (r = 0.1529, *p* = 0.029). Age was inversely correlated with eGFR (r = −0.2523, *p* < 0.001).

The present cohort was further divided into two groups by trajectory analysis: group L, with a lower eGFR (*n* = 105), and group H, with a higher eGFR (*n* = 85), as shown in Table 3 and Figure 4. Group L consisted of more men (57.14%), whereas group H included more women (80%). Compared to group H, the subjects in group L were older (*p* < 0.001), taller in BH (*p* = 0.012), heavier in BW (*p* = 0.017), and had shorter left (*p* = 0.014) and right renal lengths (*p* = 0.003). A higher proportion of subjects with an RL larger than the reference range were included in group H (*p* = 0.0197). However, over time, a linear mixed model analysis showed no significant trends of eGFR in either group L (*p* = 0.1598) or group H (*p* = 0.1456). Using survival analysis, group L could be subdivided by endpoints of eGFR ≥15 (*n* = 55) and <15 (*n* = 50), with mean follow-up periods of 1606.24 ± 1203.07 and 773.18 ± 799.33 days, respectively. Fifty subjects (47.62%) in group L, and only four subjects (4.71%) in group H, reached the endpoint KDIGO CKD stage 5 during the follow-up period (*p* < 0.001). In group L, using univariate survival analysis, a RL larger than the reference range, in comparison to a RL equal to or smaller than the reference range, was associated with a crude hazard ratio of 1.38 (95% confidence interval, 0.75–2.54) for the endpoint KDIGO CKD stage 5 without statistical significance (*p* = 0.3033); on the other hand, using multivariate survival analysis, where sex, age, BH, and BW were controlled, RL was also associated with an adjusted hazard ratio of 0.99 (95% confidence interval, 0.53–1.87) for the endpoint KDIGO CKD stage 5 without statistical significance (*p* = 0.9807).

## 4. Discussion

Ultrasonographic assessment of renal morphology is a useful and convenient method for the evaluation of almost all types of kidney diseases, including acute and chronic renal failure and structural abnormalities [13]. For many chronic renal diseases, the final renal status in common cases is a decrease in renal size related to parenchymal atrophy, sclerosis, and fibrosis [14]. The main exception is DKD, which might maintain normal kidney size, even in patients with KF [15].

In comparison to the reference range of renal length measured in a large, healthy population [7], significantly larger RL was observed in the T1DM population (*p* < 0.0001). Surprisingly, the left-sided predominance of renal length, which has been observed in the majority of studies in literature, was lost in the T1DM cohort, although statistical significance was not reached [7]. The loss of left-sided predominance might result from T1DM-related CKD, outweighing the physiological phenomenon. Renal length ultimately decreased with advanced renal failure in T1DM CKD and was inversely correlated with serum creatinine levels [9]. The renal length in T1DM patients was larger than that in non-insulin-dependent diabetes mellitus patients [9]. As seen in the general healthy population [7], the renal length was found to correlate linearly and positively with BH and BW in the T1DM population in the present study. In contrast, renal length has been shown to decrease curvilinearly with aging in the general population, reaching the maximal renal length around the age of 40, while the renal length in the T1DM population in our study was found to decrease linearly all the way with aging [7,16]. The loss of parabolically decreasing change in renal length may be attributed to the glomerular hyperfiltration in the young adulthood of T1DM and also to the T1DM-related CKD in the later years.

Although ultrasonography is a useful tool for assessing renal morphology and renal diseases, our results revealed limited application in predicting renal function decline in T1DM patients. Similarly, renal size, when initiating hemodialysis, has no predictive value in diabetic patients’ mortality rate [15], and the resistivity index using the Doppler ultrasound is not applicable for screening diabetic nephropathy in T1DM patients [17,18]. Despite the resistivity index being positively correlated with GFR in T1DM children, it has no correlation with serum creatinine and does not help predict renal function decline in adults with T1DM [19]. Increased renal size in T1DM increases eGFR via mechanisms of increased renal plasma flow, increased transglomerular hydrostatic pressure gradient, and increased glomerular ultrafiltration coefficient [8]. Histologically, mesangial, glomerular, and interstitial expansions with arteriolar hyalinosis, glomerulosclerosis, and an increased capillary diameter have been identified in T1DM-related nephropathy [20,21]. Unlike the general healthy population, the pathological changes in T1DM CKD discombobulate normal correlations between renal structure and renal function, making renal ultrasonography in T1DM an unreliable parameter for clinical use. Renal ultrasonography in T1DM is probably a much-delayed indicator of renal function, compared with serum creatinine. Since the renal dimension does not necessarily have a prognostic implication, we suggest a minor role of renal ultrasonography in the T1DM population, except for screening and monitoring renal morphological abnormalities in T1DM patients. Renal ultrasonography still has a positive clinical significance in the diagnosis of type 1 diabetic nephropathy. It may help exclude other systemic diseases, reveal the size and length of the kidney, and provide some clues in the early prediction of renal lesions, such as renal cysts, hydronephrosis, and renal stones.

In the trajectory analysis, groups L and H differed significantly in almost all baseline parameters, including age, body indices, and renal lengths. Group L, with older ages and lower eGFR compared to group H, might be viewed as a consequence of group H, rather than a comparable group parallel to group H. Despite higher and taller body indices, group L had a relatively smaller proportion of subjects, with larger renal lengths than the reference ranges, which supports their status of advanced T1DM CKD in association with poorer renal function. A larger cohort with a longer and uninterrupted follow-up is still required for a conclusive renal ultrasonographic profile of the T1DM population.

To the best of our knowledge, the present study includes the largest cohort to date and includes ultrasonography measurements of renal length in T1DM patients to elucidate the associations between renal length and age, body indices, and eGFR in the T1DM population. However, several limitations of this study must be noted. First, the number of serial measurements of an individual was limited, and the follow-up period might be relatively inadequate for a significant long-term outcome. Second, factors such as blood sugar control, amount of proteinuria, and blood pressure in type 1 diabetes that affects the deterioration of kidney function need further analysis. However, drug compliance could not be collected for analysis in the database, and since blood sugar records were missing, we did not include this part in the analysis. Third, the ultrasonographic measurement of renal length was operator-dependent, with intra- and inter-observer variability. Reliable ultrasonographic measurements require an experienced operator to identify the two-dimensional section that demonstrates the longest axis of the kidney. In the present retrospective study, renal length was measured by a large number of nephrologists from several institutions over multiple years. Therefore, the accuracy and precision of the ultrasonographic measurements in our study may be of low quality. Nevertheless, by standardizing the measurement process and method, it may be possible to reduce the magnitude of variation in kidney length measurements between individual and different nephrologists performing ultrasonographic examinations. Finally, other comorbidities, such as cardiovascular diseases, medications, or acute illnesses of any kind, were not considered in the present study, which could have an influence on eGFR and/or even renal length.

## 5. Conclusions

This study presented renal ultrasonography measurements in a large T1DM cohort and their correlations with age, body indices, and renal function, exhibiting a different renal ultrasonographic profile from the general population. Renal length measured using ultrasonography failed to predict the values for renal function decline in patients with T1DM CKD. Frequent routine renal ultrasonography examinations for T1DM have no additional benefit in monitoring known morphological abnormalities.

## Figures and Tables

**Figure 1 jcm-12-01109-f001:**
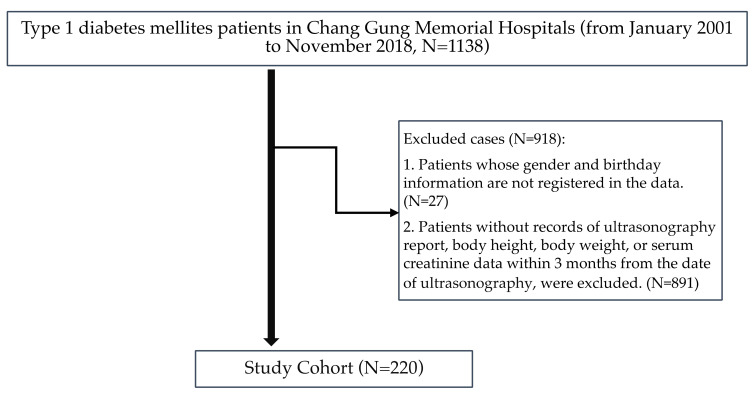
Assembly of the study cohort.

**Figure 2 jcm-12-01109-f002:**
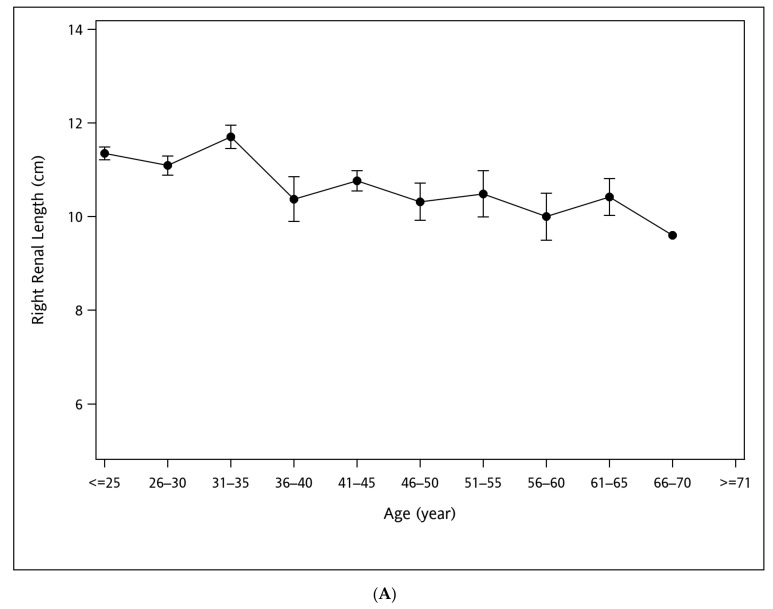
Distribution plots of right renal length at various groups of age (**A**), body height (**B**), and body weight (**C**) in the type 1 diabetes mellitus study cohort. The means of right renal length are indicated by black dots, with error bars showing 95% confidence intervals.

**Figure 3 jcm-12-01109-f003:**
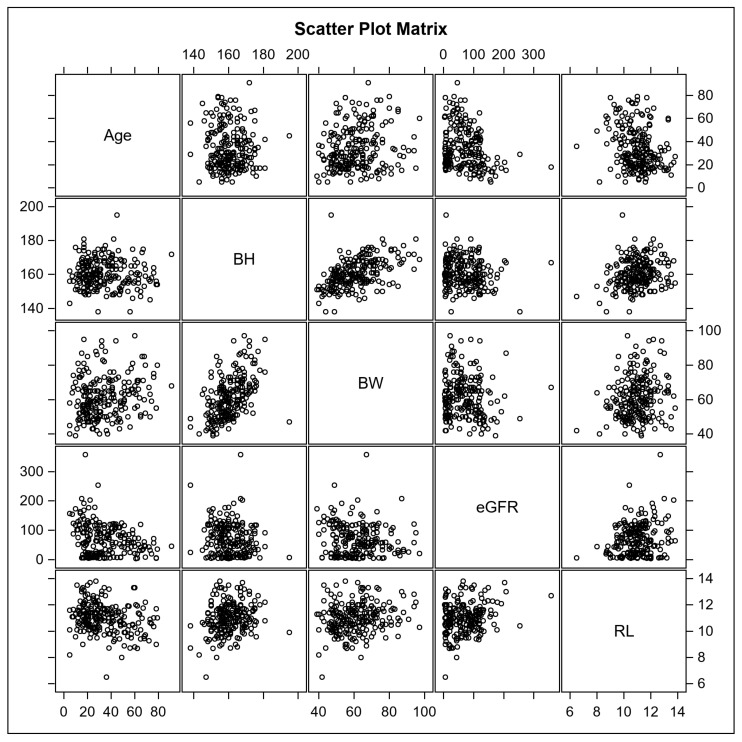
Correlations of right renal length (cm) with age (year), body height (cm), body weight (kg), and eGFR (mL/min/1.73 m^2^).

**Figure 4 jcm-12-01109-f004:**
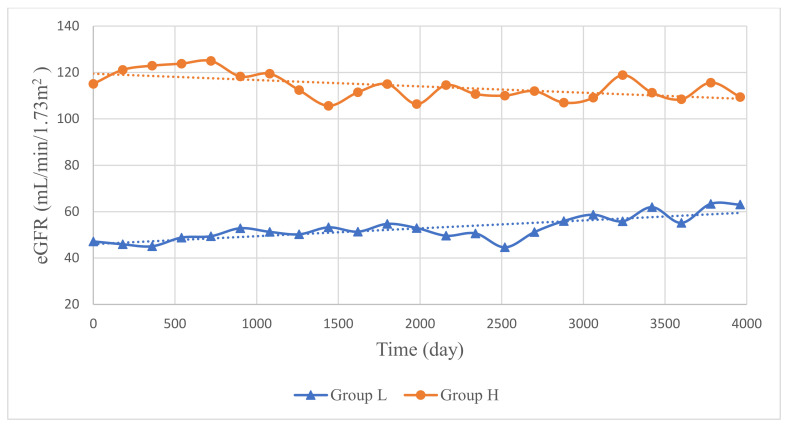
Trajectories of the estimated glomerular filtration rate (eGFR) over time in T1DM patients. Group L and group H demonstrating distinct trajectories of change in eGFR over time were delineated using the best-fitting trajectory model with the minimal Bayesian information criteria. Group H showed a decline in eGFR, while group L showed a slight incline; however, the trends of the trajectories indicated by dotted lines yielded no statistical significance. The y-axis represents eGFR (mL/min/1.73 m^2^), while the x-axis represents time (days).

**Table 1 jcm-12-01109-t001:** Demographic data of the T1DM subjects in the study cohort.

Demographics, *n =* 220	Value
Men	85 (38.6%)
Women	135 (61.4%)
Age (years)	34.2 ± 18.3
Body height (cm)	160.7 ± 8.3
Body weight (kg)	61.1 ± 11.9
BMI (kg/m^2^)	23.6 ± 3.9
eGFR (mL/min/1.73 m^2^) ^a^	60.0 (22, 84.5)
eGFR ≥ 60	75 (60.0, 112.0)
eGFR < 60	19.7 (7.8, 41.8)
Left renal length (cm) ^b^	10.9 ± 1.5
Right renal length (cm) ^c^	11.0 ± 1.1

Variables are presented as mean ± standard deviation or median (interquartile range). Abbreviation: BMI, body mass index; eGFR, estimated glomerular filtration rate. ^a^ Values reported for 148 patients. ^b^ Value reported for 219 patients. ^c^ Value reported for 218 patients.

**Table 2 jcm-12-01109-t002:** Correlation coefficients (r) of right renal length and other parameters.

		Body Height	Body Weight	eGFR	Right Renal Length
Age	r	−0.0141	0.1829	−0.2523	−0.2268
	*p*-value	0.8348	0.0065	0.0003	0.0007
Body height	r	-	0.5248	−0.1153	0.1835
	*p*-value	-	<0.0001	0.0997	0.0066
Body weight	r	-	-	−0.1063	0.1717
	*p*-value	-	-	0.1294	0.0111
eGFR	r	-	-	-	0.1529
	*p*-value	-	-	-	0.0291

**Table 3 jcm-12-01109-t003:** Comparisons between group L and group H derived from trajectory analysis.

Parameters	Group L		Group H		*p*-Value
	N/Mean	%/S.D.	N/Mean	%/S.D.	
Sex (male)	60	57.14	14	20	<0.0001 *
Sex (female)	45	42.86	68	80	
Age	39.69	19.62	28.87	14.40	0.0002 *
BH	162.33	8.62	159.24	8.31	0.0115 *
BW	63.52	11.67	59.64	12.56	0.0171 *
BMI	24.08	3.86	23.42	4.01	0.1471
LL	10.79	1.47	11.29	1.07	0.0141 *
RL	10.82	1.11	11.28	1.05	0.0037 *
RL ≤ Ref.	40	38.1	19	22.35	0.0197 *
RL > Ref.	65	61.9	66	77.65	

Abbreviations: S.D., standard deviation; BH, body height; BW, body weight; BMI, body mass index; LL, left renal length; RL, right renal length; Ref., reference ranges [7]. * Statistically significant at *p* < 0.05.

## Data Availability

The data presented in this study are available upon request from the corresponding author. The data are not publicly available because of the nature of this research; the participants of this study did not agree for their data to be shared publicly shared.

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
