# Peer review of "Ultrasonography Measurement of Renal Dimension and Its Correlation with Age, Body Indices, and eGFR in Type 1 Diabetes Mellitus Patients: Real World Data in Taiwan"

_jcm, 2023, doi:10.3390/jcm12031109_

Round 1

Reviewer 1 Report

The authors present a well-designed retrospective study and deliver a comprehensive and interesting conclusion. However, I do have some minor remarks.

- why did you choose the perdiod 2001 to 2018?
- p3, l101: what non-parametric test?
- p3, l102: I suppose the "s" is a typo?
KDIGO was never explained.
- why is the eGFR presented as median? Which test did you perform to check normal distribution? Table 1 should probably contain a detailed information whether its mean or median.
- Fig 2a: the figure, in particular the legend, can be improved. The line saying "mean" simply is the line from a point-line, and not a model, or fit, or interpolation line, is this correct?
- Fig 3: The figure contains not enough descriptions and explanations. Are the renal lengths in millimeters now? Please give some information about units.
- Fig 4 is missing units and axis titles. I in general suggest to rename "group 1" and "group 2" by comprehensive abbreviations (e.g. group low and high, or something similar; its just inconvenient to hop to the top paragraph multiple times to re-read what group meant what.

Discussion: The non-existent difference between right and left length is remarkable. The authors point out some valuable arguments, but a further argument can be the (in-)accuracy of sonographic length measurements. For those readers, who are not familiar with sonography, can you comment a little bit on this issue, please? Plus: can you assess the accuracy of the sonography length measurement, please? In general, the paragraph related to this issue is hard to read. I do understand what the authors mean, but the paragraph should be improved for the sake of readability.

Reviewer 2 Report

Comments:

1) Mention appropriate place of the study. We conducted a retrospective analysis using de-identified data retrieved from the 63 Chang Gung Research Database (CGRD).

2)  Spearman’s correlation analysis was conducted to 103 assess relationships between right renal length (RL) and renal eGFR. Give detail.

3) Make it clear. As seen in the general healthy population,[7] the 185 renal length correlates linearly and positively with BH and BW in T1DM population. In 186 contrast, the renal length in T1DM population slightly decreases with aging, while renal 187 length increases in the age of 40s and then decreases as aging in the general population.

4) In the trajectory analysis, group 1 and 2 differ significantly in almost all baseline pa- 215 rameters, including age, body indices, and renal lengths. Group 1, with older age and 216 lower eGFR, might better be viewed as a future course of group 2. Explain. 
